# Plastic and Reconstructive Surgery in the Wake of the Eid al-Adha: A Single-Center, Five-Year Investigation

**DOI:** 10.3390/jcm13092704

**Published:** 2024-05-04

**Authors:** Mehmet Tapan, Burak Yaşar, Hasan Murat Ergani, Süleyman Can Ceylan

**Affiliations:** 1Private Clinic, 07160 Antalya, Türkiye; 2Deparment of Plastic, Reconstructive and Aesthetic Surgery, Ankara City Hospital, 06800 Ankara, Türkiye; burakys@outlook.com.tr (B.Y.); dr.hasanmrt_06@hotmail.com (H.M.E.); suleymancanceylan@gmail.com (S.C.C.)

**Keywords:** Eid al-Adha, extensor tendon injuries, flexor pollicis longus tendon injury, hand surgery, maxillofacial trauma

## Abstract

**(1) Background:** The Festival of Sacrifice, commonly known as Eid al-Adha, has a profound religious and cultural impact on nations with a Muslim majority. This festival is celebrated every year in Muslim countries; however, it is a time in which patients present to the emergency department with serious injuries. In our study, we examined current injuries occurring during Eid al-Adha in one of the largest hospitals in Türkiye, providing the largest patient population to date. This included mapping tendon and maxillofacial injuries, a first in the literature. To the best of our knowledge, this is the largest case series of injuries sustained during Eid al-Adha. The significance of this study lies in its potential to significantly benefit patients and healthcare systems by providing reference data. **(2) Methods:** Patients admitted to Ankara City Hospital during Eid al-Adha between 2019 and 2023 were examined. The demographic characteristics, injury patterns, and injury sites of patients admitted on the four days of Eid al-Adha were collected and analyzed. Maxillofacial traumas during the festival were analyzed. Tendon injuries on the left hand, which is the most commonly injured body part in the literature, were mapped into a figure. To compare the change in the number of patients, a comparison was made with the number of patients in our hospital for four consecutive days 2 weeks before Eid. Statistical analysis was performed using IBM SPSS Statistics for Windows. **(3) Results:** A total of 610 patients, including 101 female and 509 male patients, were included in this study. A statistically significant increase (*p* < 0.001 for all years) in hospital admission due to injury was observed. Individuals between the ages of 30 and 40 years were the most frequently admitted patients (*n* = 182, 29.8%). Knife injuries were significantly more common in all patients (*p* < 0.001). When the total number of patients was evaluated in terms of injured areas where patients present to the emergency department, left-hand injuries were found to be significantly more common than injuries in other areas (*p* < 0.001 for all). The extensor pollicus longus tendon was the most commonly injured tendon among all extensor and flexor tendon injuries (*n* = 104). The most commonly injured tendon was the flexor tendon in zone 2 of the first finger (*n* = 45). This study showed that injuries to the extensor tendon in zone 1 of the fifth finger, the flexor tendon in zone 4 of the first finger, and the flexor tendon in zone 1 of the fifth finger were never seen. Twenty-five patients with maxillofacial injuries were admitted to the hospital. Orbital floor fractures were the most common type of maxillofacial injury. The anesthesia technique we preferred for all patients was local anesthesia (*n* = 267). Wide-awake local anesthesia no tourniquet (WALANT) was the second most preferred anesthetic technique. The number of patients who were selected in a random 4-day period for each year were compared with the number of patients who came during Eid al-Adha. The Mann–Whitney U tests revealed a significant increase in injuries on the first day of Eid al-Adha compared to non-festival days (*p* < 0.001). However, no significant differences were observed on the subsequent days or in the overall injury counts during the festival period (*p* = 0.841 for day 2, *p* = 0.151 for day 3, *p* = 0.310 for day 4). **(4) Conclusions:** According to this study, which is the largest known case series in the literature, the number of patients admitted to the hospital increased annually. In our study, we observed a significant increase in injuries only on the first day of Eid al-Adha compared to a randomly selected 4-day period of the same year. Left-hand extensor tendon injuries from a knife were the most common injuries in middle-aged men. The extensor pollicis longus tendon was the most commonly injured extensor tendon, with zones 3 and 4 being the most commonly affected. The flexor pollicis longus tendon was the most commonly injured flexor tendon in zone 2. During this period, patients may not only need hand surgery but also maxillofacial plastic surgery. We recommend, in addition to the indications I,n the literature that during Eid al-Adha, the WALANT technique should be widely adopted in patients where local anesthesia will be insufficient. We also recommend utilizing a diagram to manage the patient load during Eid al-Adha and prevent overburdening the healthcare system.

## 1. Introduction

The Festival of Sacrifice, also known as Eid al-Adha, resonates deeply in the cultural and religious fabric of Muslim-majority countries. This festival, celebrated annually in alignment with the Hajj pilgrimage, is not only a time of spiritual reflection and communal gatherings but also a period marked by a distinctive practice: the sacrificial cutting of animals, such as sheep, goats, and cattle. This ritual, rooted in the story of Prophet Abraham’s sacrifice, which is taken as an episode from the Torah and reported beyond that in the Qur’an and even in the Christian Bible, extends beyond religious observance, fostering social solidarity through the distribution of meat to family, friends, and those in need. This festival traditionally lasts for four days each year according to the lunar calendar. 

However, this auspicious event also brings forth unique medical challenges, especially in the field of plastic and reconstructive surgery [1]. A notable increase in specific types of injuries, predominantly hand and maxillofacial injuries, has been observed during this period [2]. These injuries often result from the handling and sacrifice of animals by non-professionals, using tools such as knives and cleavers without employing adequate safety measures [2,3]. This rise in cases of injury, akin to the surge observed during other festivals, such as Halloween in the West, places a significant strain on medical resources and necessitates a specialized surgical response. 

Despite the regular occurrence of these injuries during Eid al-Adha, there is a significant gap in comprehensive data on their nature, frequency, and management. This study aims to fill this gap by posing a critical question: how does Eid al-Adha affect the demand for and characteristics of plastic and reconstructive surgery services? What are the characteristics of the injuries? Can this ritual be mapped in hand injuries? In our research, we aimed to perform a five-year retrospective analysis in a single center and use our data to address these questions. 

Eid al-Adha has already made a name for itself in the literature with its association with orf infection, Crimean–Congo hemorrhagic fever, and Salmonella typhimurium infections [4,5,6]. Although the literature has focused more on hand injuries, our study is the first to comprehensively analyze maxillofacial injuries because it was conducted from the perspectives of plastic, reconstructive, and aesthetic surgery. The significance of this study lies in its potential to significantly benefit the patients and healthcare systems. To the best of our knowledge, this is the largest case series of injuries sustained during Eid al-Adha. This study also differs from other studies in that it includes the most current data, performed tendon injury zone mapping, and examined maxillofacial injuries.

## 2. Patients and Methods

In this study conducted at Ankara Bilkent City Hospital, one of the largest hospitals in Türkiye, we retrospectively analyzed data from 610 patients who presented to our Plastic, Reconstructive, and Aesthetic Surgery Clinic via the emergency department during Eid al-Adha between 2019 and 2023. The inclusion criteria were as follows: male or female patients of any age group, patients who were hospitalized or outpatients, patients with a history of injury during the four-day Eid al-Adha period consulting us in the emergency department, and/or hand injuries sustained during a procedure related to animal sacrifice; for maxillofacial injuries, a sacrifice-related story was not required, and it was considered sufficient to have been sustained during the Eid al-Adha period. Patients with inaccessible or incomplete records were excluded, as were those being treated at other hospitals. 

The analysis focused on the age and sex of the patients, etiology and date of injury, anatomical structures and locations affected, and maxillofacial trauma. Tendon injuries on the left hand, which is the most commonly injured body part according to the literature, were mapped. Anesthetic procedures including local anesthesia, wide-awake local anesthesia no tourniquet (WALANT) and general anesthesia were noted. 

To compare the change in the number of patients, a comparison was made with the number of patients in our hospital for 4 consecutive days 2 weeks before Eid. The reason why 2 weeks before the festival was selected is that Eid al-Adha falls approximately 11–12 days earlier each year compared to the previous year due to the lunar calendar. Thus, the number of patients in the previous year and the time periods coinciding with Eid al-Adha in the following year are included in the comparison group. This comprehensive inclusion of patients consulted during Eid al-Adha was vital to effectively assess the plastic surgery demand, considering that emergency cases not only included hand surgery emergencies but also traffic-accident-related injuries and maxillofacial traumas. These findings are essential for future health planning. 

This study received approval from the local ethics committee (Approval Number: E1-23-4128) and adhered to the principles of the Declaration of Helsinki.

Statistical analysis was performed using IBM SPSS Statistics for Windows, Version 23 (IBM Corp, Armonk, NY, USA). The conformity of the data to the normal distribution was visualized using histograms and probability graphs and analytical methods including the Kolmogorov–Smirnov test and Shapiro–Wilk test were applied. The descriptive statistics of parametric results were depicted by the mean (standard deviation); the descriptive statistics for nonparametric results were depicted by the median (maximum low value–high value). The independent-samples *t*-test and Mann–Whitney U test were used for parametric and nonparametric data, respectively. The ANOVA test was used to evaluate whether there was a significant difference between two variables. For nonparametric data, whether there was a significant difference between more than two variables was evaluated using the Kruskal–Wallis test. Categorical variables and the chi-square test were used to evaluate differences between two groups. A proportion z-test was used for comparisons. Statistical significance was set at *p* < 0.05.

A series of independent-samples Mann–Whitney U tests were performed to compare the number of plastic-surgery-related injuries on each day of the Eid al-Adha festival to the injuries on non-festival days for each corresponding day across five years. The tests aimed to assess whether there was a statistically significant difference in injury counts between festival (Eid) and non-festival (Random) days. The significance level was set at *p* < 0.05 for all tests.

## 3. Results

A total of 610 patients, including 101 female and 509 male patients, were included in the study (Table 1). When all patient numbers were evaluated on an annual basis, there was no significant difference between 2019 and 2020 (*p* = 0.260), while there was a statistically significant increase in the number of patients in the other years compared to 2019 (*p* < 0.001 for all years). Compared with 2020, the number of patients increased significantly in 2021, 2022, and 2023 (*p* < 0.001, 0.001, and 0.015, respectively). In 2021, the highest number of patients was observed, but there was no statistical difference between 2022 and 2023 (*p* values = 0.136 and 0.548, respectively).

In the last five-year period, 90.6% of patients admitted on the first day were male, while this rate was 81.8% on the second day, 69.7% on the day day, and 69% on the fourth day. The proportion of men who visited the hospital on the first day was significantly higher than that of those who visited the hospital on the other days (*p*-values: 0.028, <0.001, <0.001, respectively). When all days were evaluated individually, the number of males was significantly higher than that of females (*p* < 0.001 for all days). Analyzing the total male/female ratio by years, it was 77.4% in 2019, 88.8% in 2020, 84.3% in 2021, 82.6% in 2022, and 83.2% in 2023. When the change in the male-to-female ratio was evaluated over the years, no statistically significant difference was found (*p* = 0.354) (Table 1). 

Individuals between the ages of 30 and 40 years were the most frequently admitted patients (n = 182, 29.8%) (Table 2). This was followed by individuals aged 40–50 years at a rate of 26.2% (n = 160). The lowest rate was found in individuals aged 80–90 with 0.82%. The admission rate of individuals aged 70–80 was the second lowest at 1.15%. There was no significant difference between individuals aged 30–40 and 40–50 (*p* = 0.160). However, individuals in these age groups (30–40 and 40–50 years) had significantly higher admission rates than those in all other age groups, which were 0–10 (*n* = 11), 10–20 (*n* = 39), 20–30 (*n* = 75), 50–60 (*n* = 82), 60–70 (*n* = 49), 70–80 (*n* = 7), and 80–90 (*n* = 5) (all *p* < 0.001). When changes in the age group distribution over the years were examined, no significant changes were observed (*p* = 0.231).

Comparing injuries from other injury types to those caused by knives, we discovered that knife injuries ranked highest across all of the time periods. This was significantly higher than all of the other injuries (*p* < 0.001). Thus, knife injuries were significantly more common in all patients (*p* < 0.001) (Table 3).

When the total number of patients was evaluated in terms of the areas injured when patients presented to the emergency department, left-hand injuries were found to be significantly more common than injuries in other areas (for all *p* < 0.001). When we examined the years individually, the left hand was injured at significantly higher rates than all other injury sites in 2019, 2020, and 2023 (for all *p* < 0.001). In 2022, there was no significant difference between left- and right-hand injuries (*p* = 0.149), but they were both significantly higher than all of other injury sites (for all *p* < 0.001) (Table 4). 

The examination of the years individually revealed that extensor tendon laceration was significantly more common than other types of injury (*p* < 0.001 for all) (Table 5). The extensor pollicus longus (EPL) tendon was the most injured tendon among all extensor and flexor tendon injuries. The number of patients whose EPL tendon was injured alone was *n* = 77, and the number of patients whose EPL tendon was injured together with other extensor tendons was *n* = 27. The number of patients whose flexor pollicis longus (FPL) tendon was injured was *n* = 63. The most injured tendon was the flexor tendon in zone 2 of the first finger (*n* = 45). In terms of tendon injuries, the first finger tendons were the most commonly injured, either alone or with other fingers (*n* = 167). There were no tendon injuries to the extensor tendon in zone 1 of the fifth finger, the flexor tendon in zone 4 of the first finger, or the flexor tendon in zone 1 of the fifth finger. 

A map of the extensor and flexor tendon injury sites on the left hand is shown in Figure 1. It shows only the isolated finger injuries. The figure does not show twenty-three patients (7.4%) with injuries to both the first and second finger extensor tendons, four patients (1.2%) with injuries to the first, second, and third finger extensor tendons, seven patients (2.2%) with injuries to the second, third, and fourth finger extensor tendons, and eight patients (2.5%) with injuries to both the fourth and fifth finger extensor tendons. In addition, sixteen patients (8.8%) with flexor tendon zone injuries to both the second and third finger flexor tendons are not shown on the map. 

All tendons and skin were repaired with a polyprolene material. Flexor tendons were repaired with a 3-0 polyprolene core suture with a modified Kessler technique and a 6-0 polyprolene epitendinous suture. Extensor tendons were repaired with sutures ranging from 3-0 to 5-0 polyprolene depending on the size of the tendon.

Patients with maxillofacial trauma are listed in Table 6. Orbital floor fractures were the most common type of maxillofacial injuries.

The anesthesia technique we preferred for the patients was mostly local anesthesia (*n* = 267). The number of patients administered WALANT anesthesia was 227 and the number of patients given general anesthesia was 96.

The number of patients who were selected in a random 4-day period are shown in Table 7. The Mann–Whitney U tests revealed a significant increase in injuries on the first day of Eid al-Adha compared to non-festival days (*p* < 0.001). However, no significant differences were observed on the subsequent days or in the overall injury counts during the festival period (*p* = 0.841 for day 2, *p* = 0.151 for day 3, *p* = 0.310 for day 4). These results suggest a possible association between the commencement of Eid al-Adha and an increased number of injuries, but such a pattern does not persist throughout the festival.

## 4. Discussion

Injuries or other diseases can have terrible consequences and change the mood of peaceful festivals into agony. For instance, during Halloween and Easter in Canada, an increased risk of anaphylaxis is induced by unknown nuts and peanuts lurking in the treats handed out [7]. Notably, Eid al-Adha is another such festival period marked by unexpected medical issues. Based on the number of patients admitted to our hospital, the increasing trend in the number of injuries continues to be statistically significant. The female-to-male ratio in our study was 1:7, and it is approximately 1:4 in the literature [8,9]. Similar to the literature, we observed that the proportion of males ranged from 77.2 to 88.8%.

Hand injuries commonly occur during the sacrificing and processing of meat, and reports of injuries in patients ranging from 3 to 67 years of age have been published in the literature [8,10]. Only the first three days of the four-day sacrifice feast are available for sacrifice. As this is a traditional act, men often carry out the sacrifice. Women typically perform meat slicing. In our study, men were admitted to the hospital on the first day at a significantly higher rate than men who presented on subsequent days. However, upon analyzing each day separately, we found that there were still significantly more men than women. We also found that there were significantly higher numbers of individuals in the 30–40 and 40–50 years age ranges than in any other age group. No change was observed in the age group distribution over the years. This was also expected due to traditional behavior.

In our study, extensor tendon injury was the most common type of injury, which was consistent with the literature, in which it varied from 47 to 68.9% [11,12]. We also found that a left-hand injury with a knife was the most common anatomical location and mechanism (Figure 2). Fortunately, meat mincer injuries, which cause more serious trauma such as amputations, were rarer than knife injuries. However, injury with a sharp instrument such as a knife causes exposure to injuries below skin level. The process of slaughtering the animal and then slicing the meat explains the high incidence of knife injuries. Since this process will continue throughout the four days of the festival, the incidence of knife injuries is high for each day. It is also believed that the nondominant hand sustains more cuts from knives [13].

Our mapping in Figure 1 clearly shows that the left-hand dorsum extensor tendon in zones 3 and 4 of the first finger are the most commonly injured extensor tendon zone areas. In isolated finger injuries, the most common injury was a flexor tendon injury in zone 2 of the first finger. The map also shows that injuries to the extensor tendon in zone 1 of the fifth finger, the flexor tendon in zone 4 of the first finger, and the flexor tendon in zone 1 of the fifth finger were never seen. The mapping shows that the first finger was the finger most affected by tendon injuries while the fourth finger was the least affected. These data not only constitute the first tendon zone mapping of injuries during this festival in the literature but also the first mapping of hand injuries in the literature.

As plastic surgeons, we observed that there were not only hand injuries but also maxillofacial traumas. During the Eid al-Adha period, the number of people traveling between cities increases, which leads to an increase in traffic accidents. In the literature, it was previously reported that being on vacation plays a role in crash severity [14]. A total of 25 patients with maxillofacial trauma were enrolled in our study. Notably, 40% of the patients had orbital floor fractures, which represent the most common injury type (Figure 3). This fracture was caused by the impact of an injuring agent incapable of transitory deformation, such as traffic accidents, or the body part of an animal, which we encountered in two patients. Two out of thirty-six cases of animal kicks resulted in orbital fractures, which is a significant number and something that emergency services should take into consideration during this festival. 

As a surgical indication, diplopia caused by compression of the inferior rectus muscle and airway problems due to maxillofacial fracture should always be kept in mind as they require surgery within a short time period. Additionally, given that sacrifices are traditionally made during the first three days of the four-day festival, these injuries can be expected during the first three days. In four patients, the maxillofacial trauma type was a mandibular fracture due to traffic accidents.

Only 96 of 610 patients required general anesthesia. Local anesthesia and the WALANT anesthesia technique, which has become widespread in recent years, can provide a rapid solution to the patient load. WALANT can be used successfully for most common hand procedures [15]. WALANT is a technique using local anesthetic and hemostatic agents to recover from conditions without using a tourniquet and sedation for hand surgery. One of the indications of WALANT includes limited access to healthcare. Ambulatory surgical care can be provided in rural or underdeveloped areas without the requirement for an operating room or an anesthetic team [16,17]. WALANT has been shown to be consistently cheaper and more efficient than the operating room when it is applicable [18]. We think it is a good choice because it provides a quick solution in situations in which many patients arrive at once. We recommend, in addition to the indications in the literature, that the WALANT technique be widely adopted during Eid al-Adha, for patients where local anesthesia would be insufficient. 

In our study, we observed a significant increase in injuries only on the first day of Eid al-Adha compared to a randomly selected 4-day period of the same year. This is proof that there is indeed an increase in injuries during Eid al-Adha. We think that the reason why there was no significant increase in the number of patients on the second, third, and fourth days of Eid al-Adha was that the workers who were on vacation during the festival did not have hand injuries, with work accidents occurring on the randomly selected days.

One of the limitations of this study is that it was conducted in a single hospital. However, Ankara Bilkent City Hospital is one of the largest hospitals in a country with a population of over 80 million. Additionally, the educational status of patients was not included in the study. Although we extensively researched injury sites and anesthesia techniques, we did not include the treatment and long-term outcomes. Although our mapping may guide reoperative interventions, we do not have any data on these.

Considering that this festival will continue in the future, it may make sense to develop protective equipment based on the injury results of our mapping. In addition, in our study, we drew attention to the fact that Eid al-Adha is a period in which maxillofacial injuries may also occur, in contrast to the perception that it is a period when only hand injuries increase. A diagram summarizing the timing of the treatment of these injuries during Eid al-Adha is shown in Figure 4.

## 5. Conclusions

During Eid al-Adha, specialists with sacrificial cutting qualifications should perform the sacrifice. As knife injuries are very common, we recommend that this procedure be carried out by professionals or that special protective equipment be developed. The scientific particularity of this study is not only found in its quantitative abundance but also in the detailing of the tendon injuries in particular. Left-hand knife injuries are very common during Eid al-Adha. The EPL tendon was the most commonly injured extensor tendon, with zones 3 and 4 being the most commonly affected. The FPL tendon was the most commonly injured flexor tendon in zone 2. According to our large case series, during this period, patients may not only need hand surgery but also maxillofacial surgery, including plastic surgery. Hospital emergency departments, especially on the first day of Eid al-Adha, must be adequately staffed and equipped to deal with the increasing number of injuries that occur during Eid al-Adha. We recommend utilizing a diagram to manage the patient load during Eid al-Adha and prevent overburdening the healthcare system.

## Figures and Tables

**Figure 1 jcm-13-02704-f001:**
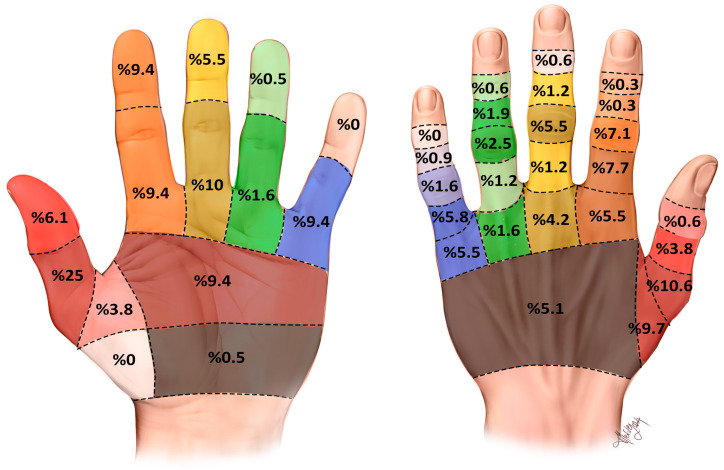
The map of left-hand tendon lacerations isolated for each finger. The figure on the left shows the numbers of patients who had flexor tendon injuries according to flexor tendon zones. The figure on the right shows the numbers of patients who had extensor tendon injuries according to extensor tendon zones. Each finger was painted in its own color.

**Figure 2 jcm-13-02704-f002:**
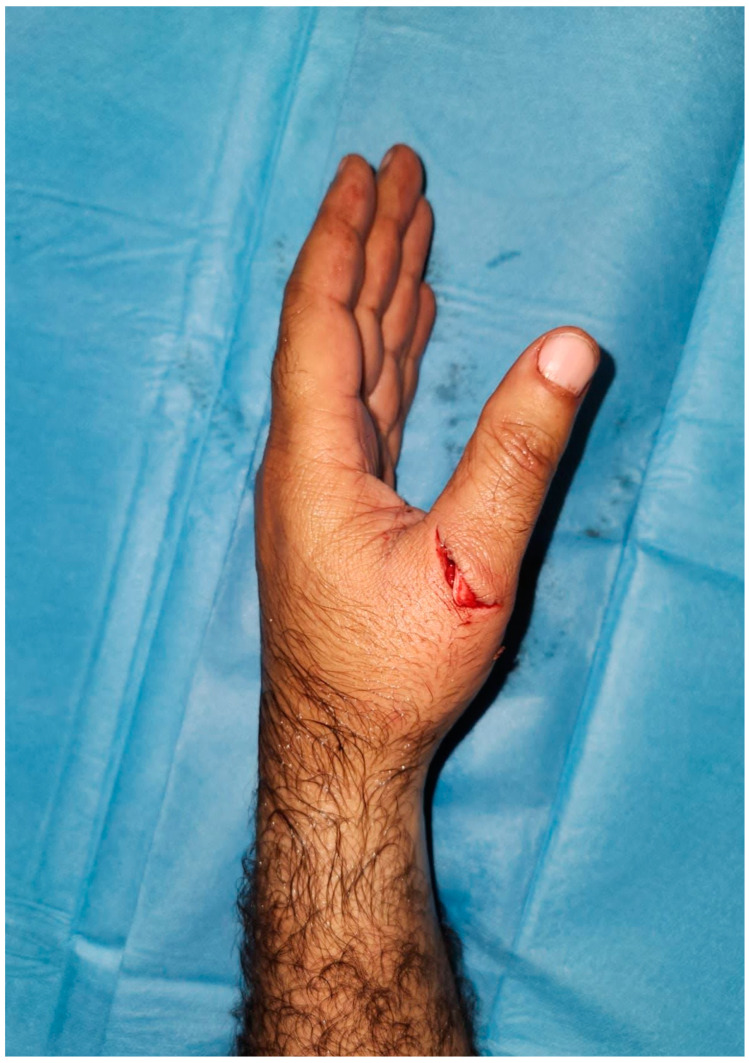
A 35-year-old male who suffered from a knife injury to his extensor pollicis longus tendon. This case belonged to the most common patient and injury type.

**Figure 3 jcm-13-02704-f003:**
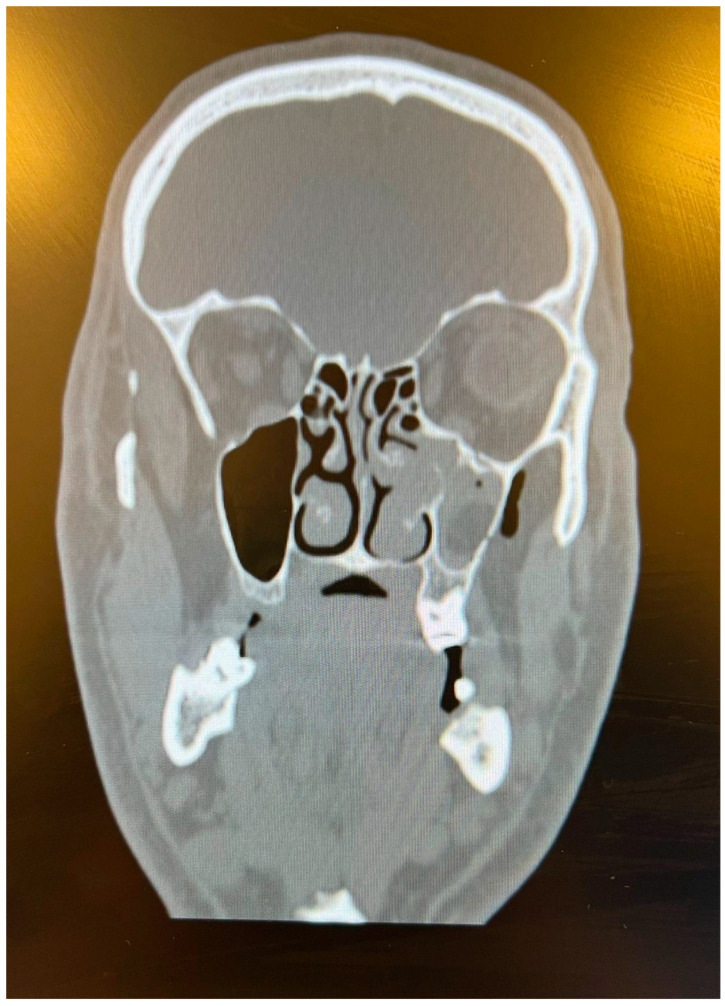
A 38-year-old male who suffered an animal kick to his face. The CT scan shows a left-sided orbital floor fracture.

**Figure 4 jcm-13-02704-f004:**
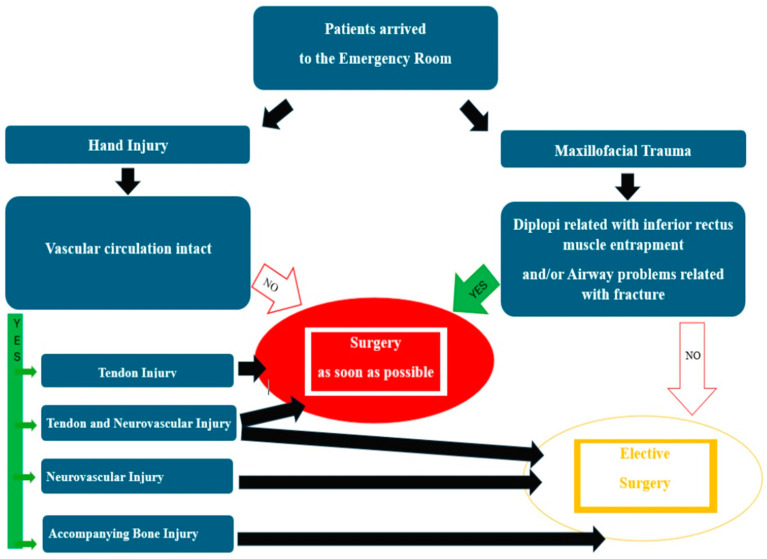
A summary of the options for managing injuries in the department of plastic, reconstructive and aesthetic surgery. Vascular circulation problems of the hand, diplopia related with inferior rectus muscle entrapment and airways problems of maxillofacial trauma are already a surgical emergency. Tendon injury and tendon and neurovascular injuries can be managed in the emergency setting. During this festival, the accumulation of patients can be prevented by operating on such short cases.

**Table 1 jcm-13-02704-t001:** The number of the patients including their sex.

	Number of Patients Female/Male (Total %)				Total (%)
Year	1st day	2nd day	3rd day	4th day	
2019	6/51 (67.8%)	1/9 (11.9%)	7/2 (10.7%)	5/3 (9.5%)	84 (13.8%)
2020	3/58 (62.2%)	5/15 (20.4%)	2/9 (11.2%)	1/5 (6.1%)	98 (16%)
2021	9/62 (46.4%)	5/27 (20.9%)	6/23 (18.9%)	4/17 (13.7%)	153 (25.1%)
2022	8/69 (53.4%)	4/17 (14.5%)	5/16 (14.5%)	8/17 (17.3%)	144 (23.6%)
2023	6/68 (56.4%)	5/22 (20.6%)	7/12 (14.5%)	4/7 (8.3%)	131 (21.5%)
Total (%)	340 (55.7%)	110 (18%)	89 (14.6%)	71 (11.7)	610% 100

**Table 2 jcm-13-02704-t002:** The age groups of the patients.

Age Group	2019 (*n*)	2020 (*n*)	2021 (*n*)	2022 (*n*)	2023 (*n*)
0–10	2	1	0	0	2
10–20	4	6	10	8	11
20–30	10	12	15	17	21
30–40	23	22	47	41	49
40–50	22	28	49	35	26
50–60	14	21	20	17	10
60–70	7	5	10	15	12
70–80	1	2	1	3	0
80–90	1	1	1	2	0

**Table 3 jcm-13-02704-t003:** The types of injuries in the patients.

Injury Type	2019 (*n*)	2020 (*n*)	2021 (*n*)	2022 (*n*)	2023 (*n*)	
Knife	43	53	85	74	77	332
Animal kick	2	3	12	11	8	36
Meat mincer	3	4	8	5	7	27
Fall	11	5	10	15	9	50
Entanglement in a chain	2	1	3	9	6	21
Other	12	22	19	14	21	88
Traffic accident	8	10	12	15	3	48
Human bite	3	0	4	1	0	8

**Table 4 jcm-13-02704-t004:** The location of the patients’ injuries.

	2019 (*n*)	2020 (*n*)	2021 (*n*)	2022 (*n*)	2023 (*n*)	
Left hand	60	70	76	65	73	344
Right hand	13	12	49	53	36	163
Right lower extremity	2	3	8	8	6	27
Left lower extremity	2	1	9	5	5	22
Neck	0	2	3	0	0	5
Abdomen	1	2	0	1	0	4
Head	2	3	3	2	1	11
Chest	4	5	5	10	10	34

**Table 5 jcm-13-02704-t005:** The injured body parts of the patients.

	2019 (*n*)	2020 (*n*)	2021 (*n*)	2022 (*n*)	2023 (*n*)	
Only skin laceration	12	15	26	22	32	107
Flexor tendon injury	12	20	31	31	22	116
Extansor tendon injury	50	54	75	69	61	309
Flexor tendon injury with nerve and vessel injury	5	12	15	15	17	64
Fingertip amputation	4	3	7	7	5	26
Finger amputation	3	2	4	7	2	18
Hand amputation	1	0	1	2	1	5
Maxillofacial trauma	2	4	5	6	8	25

**Table 6 jcm-13-02704-t006:** The injured maxillofacial trauma of the patients.

Zone of Maxillofacial Injury	2019 (*n*)	2020 (*n*)	2021 (*n*)	2022 (*n*)	2023 (*n*)
Orbital floor	1	1	2	2	4
Zygoma	-	1	2	1	1
Maxilla	1	1	1	1	2
Mandible	-	1	-	2	1

**Table 7 jcm-13-02704-t007:** The patients who had hand injury and went to our emergency service in random 4-day period of each year.

Year	Number of Patients				
	Day 1	Day 2	Day 3	Day 4	Total
2019	9	16	22	17	64
2020	18	10	19	22	69
2021	20	17	27	20	84
2022	23	30	25	24	102
2023	33	43	34	29	139

## Data Availability

The data underlying this article will be shared on reasonable request to the corresponding author.

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
