# Peer review of "Plastic and Reconstructive Surgery in the Wake of the Eid al-Adha: A Single-Center, Five-Year Investigation"

_jcm, 2024, doi:10.3390/jcm13092704_

Round 1

Reviewer 1 Report

Comments and Suggestions for Authors

Dear Authors,

This observational study has the prospective to be used as a guidance for better health management during this particular time period. However, there are some points needing further clarification. It is important to become clear between which periods were made the index comparisons and statistical analysis. Moreover, WALANT technique should be described in short and also the recommendation of this technique should be justified.

Lines 77-78: Is there any possibility patients fulfilling inclusion criteria not to be admitted in Plastic Surgery Department but treated in Emergency Department (ED) either during the examining period or the rest of the year?

Lines 109-112: Several comparisons were made but it is unclear the meaning of them. As far as I am concerned, each Eid al-Adha period should be compared with the rest of each year respectively  or  any random 4-day period  in order concrete and valuable conclusions to be drawn. 

Line 176: The discussion section should be formed according to new results as it was described above. Moreover, in this section relevant literature outcomes are compared with the results of the index study. 

Lines 250-255 and lines 261-262: These sentences are totally irrelevant with the rest of the manuscript. In conclusion section, results of the study are summarised in a succint and clear fashion without adding any new information which is not mention throughout the whole manuscript.

Comments on the Quality of English Language

Dear authors,

There are several points needing editing as comprehension of the meaning is impossible.

Reviewer 2 Report

Comments and Suggestions for Authors

Overall this is a well designed and well conducted study.

Some comments:

1. Provide some data about the incidence and causes of injures

2. Please provide inclusion and exclusion criteria in methods.

3. Please provide some details to the type of reconstructive surgery.

4. Please make a summary of the options to manage these injures.

Comments on the Quality of English Language

Minor editing of English language are required.

Reviewer 3 Report

Comments and Suggestions for Authors

Dear Authors, 

The subject of the manuscript is interesting especially as an etiological context.

The number of treated cases is considerable.

The article is, in my opinion, only a statistical study of some hand injuries in a special etiological context.

The scientific particularity of the study is not emphasized by the authors.

Perhaps some particularities of the injuries produced in the special context described should be emphasized

Best regards

Round 2

Reviewer 1 Report

Comments and Suggestions for Authors

Dear Authors, 

Significant improvement has been made in this revised version of the manuscript. I would like to mention two points needing further clarification and improvement. Please justify the rationale according to which "no significant differences were observed ...(line 30)" with p>0.001. Level of statistical significance could be p<0.001, p<0.01 or p<0.05. You have to choose and defend in a scientific fashion this choice of yours. The same applies to lines 208-212. Secondly, please improve the presentation of figure 4 diagram. Last but not least, as I mentioned before (round 1 of review) lines 250-255 then (323-327 now) are totally irrelevant to the content of manuscript's main part. 

Reviewer 3 Report

Comments and Suggestions for Authors

Dear Authors, 

Congratulations for your work!

Best regards

Author Response

We highly appreciate the detailed valuable comments of the referees on our manuscript of jcm-2970517. 

Thanks and Best Regards

 Yours Sincerely,

Mehmet Tapan

2024-04-30